# Psychological Well-Being and Home Conditions during COVID-19 Confinement. Internet Addiction and Nostalgia as Mediators

**DOI:** 10.3390/ijerph18147386

**Published:** 2021-07-10

**Authors:** Mario Del Líbano, Miguel Corbí, Aida Gutiérrez-García, Almudena Alonso-Centeno

**Affiliations:** 1Departamento de Ciencias de la Educación, Facultad de Educación, Universidad de Burgos, 09001 Burgos, Spain; mdlibano@ubu.es; 2Departamento de Didácticas Específicas, Facultad de Educación, Universidad de Burgos, 09001 Burgos, Spain; aacenteno@ubu.es; 3Departamento de Ciencias de la Salud, Facultad de Ciencias de la Salud, Universidad de Burgos, 09001 Burgos, Spain; aidagg@ubu.es

**Keywords:** home conditions, internet addiction, nostalgia, psychological well-being, confinement, COVID-19

## Abstract

The COVID-19 pandemic posed a challenge for all confined populations, dealing with their home resources and suffering changes in their psychological well-being. The aim of this paper is to analyze the relationship between home conditions (i.e., having children, square meters of the house and square meters of the terrace or similar) and psychological well-being, and to test whether this relationship is mediated by Internet addiction and nostalgia. The sample was composed of 1509 people, aged between 18 to 78 years (67.6% women). Structural Equations Models and 2 × 2 ANOVAs were analyzed. It was found that better home conditions mean greater psychological well-being, and that this relationship is partially mediated, in a negative sense, by Internet addiction and nostalgia, especially after day 45 of confinement and with greater intensity in women. These results provide evidence about how psychological well-being can be preserved during a confinement situation, which may be useful for planning healthy strategies in similar circumstances in the future.

## 1. Introduction

One of the most notable events of 2020 is undoubtedly the global pandemic due to the spread of the virus called SARS-CoV2 (Severe Acute Respiratory Syndrome Coronavirus 2), which gave rise to the disease known as COVID-19 (COronaVIrus Disease 2019) [1]. This fact led the governments of different countries around the world to enforce restrictions to contain the spread of the virus, even going as far as exceptional measures such as quarantine. Other countries only recommended that the population be kept as isolated as possible. Both isolation and quarantine have proven successful in the past in restoring public health [2], although they can also evoke controversial emotions such as fear, resentment, and perplexity [3,4]. In this context, Spain was one of the first countries affected in the world. In sight of the increasing number of infected cases, the Spanish government decreed a “state of emergency” on March 15, which lasted until June 21, and maintained a quarantine framework from the beginning of the state of emergency until May 2nd. Thus, 48 of the 98 days of confinement were requested to the Spanish population to remain in quarantine circumstances, except for situations such as the purchase of food and medicine, work issues or emergencies. This measure also included the closure of educational and business centers, and non-essential factories and stores such as shopping centers, restaurants, and coffee shops [5], which lead to conditions of social alienation and movement limitation [6].

While quarantine has never been seen before by most of the world’s population, these measures were used several times throughout history with the same objective as today. Quarantine is the period of isolation imposed on people, animals or things that can spread a contagious disease [3,7]. It is characterized by a period in which people restrict their social relations and impose limits on their movement [7,8], changing habits. These changes may be influenced by home conditions [9,10], which can also affect people’s psychological well-being (PWB) [11,12]. The current technological development makes a big difference between the closure of COVID-19 and the previous isolation times. In this sense, some studies suggest that the progress of technology, even before the pandemic’s confinement, caused people to prefer online over face-to-face channels for purposes of communication, learning social rules, recreation, and entertainment, etc. [13,14]. Thus, the wide range of possibilities for communicating online offers many alternatives for people to keep in touch with their loved ones.

Despite the advantages that Internet offers through, for example, online resources, its excessive and compulsive use can lead some people to develop Internet addiction (IA). IA has been proposed as an explanation for the uncontrollable and harmful use of technology to access the Internet. It has been characterized as an impulse control disorder comparable to pathological gambling because of overlapping diagnostic criteria and symptomatology [15]. Many studies have looked at how IA influences people’s PWB, and most have found that has a negative effect [16,17]. There is also evidence that Internet abuse leads to higher levels of depression, a greater feeling of loneliness and stress [18,19].

The deprivation of movement inherent to confinement is a proper situation to emerge feelings of nostalgia in people, characterized by sentimental longing for other and personal events from the past [20]. Although some research has shown nostalgia has a positive influence on PWB [21,22], when it is experienced in threatening or unpleasant situations, it can be expected to have a negative effect [23]. The constant hail of negative news and the information about COVID-19 could vastly increase the feeling of nostalgia and activate thoughts of loneliness [20,24], meaninglessness [22] and death [25,26].

People change throughout their lives based on their experiences. These changes can be greater when the experiences are more intense. Therefore, it is to be expected that during COVID-19 confinement, many people changed some of their behaviors, and abandoned activities that they used to do. Nostalgia may also be significantly influenced by containment relief measures, as this is a good time to remember the activities that have been missed and plan when they will be undertaken again. Therefore, in the case of Spain, we can expect differences in behavior from the moment the first week of relief was announced, that is, between the 44th and 45th days of quarantine. 

In accordance with the studies aforementioned, the general objective of this paper is to determine if home conditions (i.e., having children, square meters of the house and square meters of the terrace or similar) relate to the PWB of people in confinement and if this relationship is mediated by IA and by nostalgia.

The hypotheses arising from this objective are as follows:

**Hypothesis** **1** **(H1).**
*There will be a partial mediation of IA between home conditions and PWB.*


**Hypothesis** **2** **(H2).**
*There will be a partial mediation of nostalgia between home conditions and PWB.*


**Hypothesis** **3** **(H3).***The partial mediation of IA will be greater from the 45th day of confinement*.

**Hypothesis** **4** **(H4).**
*The partial mediation of nostalgia will be greater from the 45th day of confinement.*


**Hypothesis** **5** **(H5).**
*There will be an interaction effect between sex and days of confinement in IA and nostalgia.*


## 2. Materials and Methods 

### 2.1. Sample

The sample is composed of 1509 people of different ages, genders and origins (1020 women, 67.6%), who met the following inclusion criteria: (1) to stay in Spain at the time of confinement; (2) to be at least 18 years old (age at which it is legally possible to work in Spain); and (3) to accept participation in the study in the online questionnaire. In addition, we established an exclusion criterion, that is, not having completed all the questions in the questionnaire (except for those that were not mandatory).

The average age of the participants was 37.23 years old (Standard Deviation, SD = 13.91), ranging from 18 to 78. The main situations in which people found themselves in terms of their occupation were: 32.8% worked online, 25.6% were students, 12.1% were in a temporary employment regulation file (ERTE in Spanish), 11.5% were unemployed and 9.7% worked face-to-face. Moreover, 29.5% had at least one child. The average number of days of confinement was 45 days (SD = 3), ranging from 33 to 68. Only 6.8% had to be isolated due to COVID-19. 

### 2.2. Procedure

A quantitative research approach with a cross-sectional design was adopted in this study. The study was approved by the local bioethics committee of the University of Burgos (IR32/2020), in accordance with the World Medical Association Helsinki Declaration 2008. The online software provided by www.onlineencuesta.es (accessed on 13 June 2021) was used to create and distribute the questionnaire. The questionnaires were administered online between 26 April and 22 May 2020 and disseminated through various news websites and social networks such as LinkedIn, which reported the start of research into the effects of confinement on people’s well-being.

The first page of the questionnaire contained information on the anonymity and confidentiality of the responses, as well as a request for consent for processing. In addition, information was provided on the purpose of the research, the persons targeted, the duration of the questionnaire, and the data protection regulations on which it was based. 

### 2.3. Measures

#### 2.3.1. Sociodemographic Data

Information was collected on sex (1 = man, 2 = woman), age, time of confinement in days (which was later grouped into 1 = up to 44 days of confinement and 2 = from 45 days of confinement), type of occupation, and time in isolation in the case of being positive for COVID-19.

#### 2.3.2. Home Conditions

The number of children, the square meters of the house and the square meters of the terrace, balcony or similar, were asked through open questions.

#### 2.3.3. Nostalgia

To assess nostalgia, we created an ad hoc questionnaire consisting of 13 items that were answered according to a scale of four alternatives comprising 1 = “I don’t miss it at all” and 4 = “I miss it a lot”. The items were created following the guidelines established by [27]: each item should address only a single issue; items should be clear, simple, and concise; items with repeated information should be avoided; the vocabulary should be accessible to all members of the population; and biased items should be avoided. Specifically, participants had to indicate how much they missed 13 different activities. The higher the score, the more nostalgia (α = 0.74). The 13 items were classified into 3 categories according to an exploratory factor analysis computed with SPSS: (1) leisure-time activities, which includes 7 items (e.g., “How much do you miss going to restaurants?”); (2) intellectual activities, which includes 3 items (e.g., “How much do you miss going to libraries?”); and (3) basic activities, which includes 3 items (e.g., “How much do you miss spending time with the family?”). 

#### 2.3.4. Internet Addiction Test (IAT; Young, 1998)

The Spanish version of IAT [28] was used to measure the level of IA. The IAT comprises 20 items that inquire about habits and patterns of Internet use with five response choices from 1 = “rarely” to 5 = “always”. The minimum score is 20 and the maximum is 100. Scores from 20 to 49 indicate that there is control over Internet use, 50 to 79 frequent problems with Internet use, and a score above 80 refers to significant problems in life due to Internet use. An example of an item is “How often do you try to reduce the time you spend on the Internet and fail to do so?”. Because the questionnaire was answered during COVID-19 confinement, one of the items was not analyzed to avoid distortions in the total score. This item was: “How often do you prefer to spend more time on the Internet than dating?” The higher the score, the greater the IA (α = 0.89).

#### 2.3.5. General Health Questionnaire (GHQ-12; Goldberg and Williams, 1988)

The Spanish version of GHQ-12 [29] was used to measure PWB. The questionnaire comprises 12 items that assess psychological health problems over the past few weeks using a 4-point scale. Specifically, each item is accompanied by four possible responses from 1 = “not at all” to 4 = “much more than usual”. An example of an item is “Have you recently been feeling reasonably happy, all things considered?” The higher the score, the better the PWB (α = 0.88).

### 2.4. Data Analysis

Firstly, we computed the internal consistencies, means, SDs and correlations. Second, to test the relationship among home conditions, IA, nostalgia, and PWB, Structural Equation Models (SEM) were calculated using AMOS software (Version 25, IBM Corp, Armonk, NY, USA). Third, to explore possible differences in results between days of confinement, MuLti-Group (MLG) analyses were performed. This technique looks for statistically significant differences in trajectory coefficients between sub-samples [30]. Finally, to analyze interaction effects of sex and days of confinement in IA and nostalgia, 2 (sex) x 2 (confinement) ANOVAs were conducted. Effect sizes (ŋ^2 partial^) for main effects and interactions were included.

We used maximum likelihood estimation methods, and the input for each analysis was the covariance matrix of the variables. We tested different fit indices: the ¦Ö2 Goodness-of-Fit Statistic, Goodness-of-Fit Index (GFI), Adjusted Goodness-of-Fit Index (AGFI), the Root Mean Square Error of Approximation (RMSEA), the Tucker–Lewis Index (TLI), the Comparative Fit Index (CFI), the Normed Fit Index (NFI), and the Incremental Fit Index (IFI). According to Browne and Cudeck (1992) values smaller than 0.05 for RMSEA indicate a good fit and values between 0.05 and 0.08 indicate an acceptable fit. For the remaining indices, values greater than 0.90 indicate a good fit. A revised cut-off value of CFI close to 0.95 was also advised [31]. 

## 3. Results

The mean values, standard deviations, and intercorrelations among study variables are shown in Table 1. On the one hand, the different dimensions of nostalgia correlate with each other. On the other hand, IA is negatively related to PWB, which in turn is negatively related to nostalgia and positively related to number of children. 

Furthermore, the results of Harman’s single factor test with CFA [32] reveal a significantly lower fit to the data, χ^2^_(54)_ = 1605.07, GFI = 0.83, AGFI = 0.76, RMSEA = 0.14, TLI = 0.53, CFI = 0.62, NFI = 0.62, IFI = 0.62, so one single factor could not account for the variance in the data. Consequently, the common method variance is not a deficiency in the dataset.

### 3.1. Model Fit: Testing the Hypotheses

To contrast H1 and H2 an SEM was performed. The model fits the data well (χ^2^_(48)_ = 232.72, GFI = 0.97, AGFI = 0.96, RMSEA = 0.05, TLI = 0.94, CFI = 0.95, NFI = 0.94, IFI = 0.95). As can be seen in Figure 1, home conditions related significantly and positively to PWB. This relationship was partially mediated by IA (which was negatively related to home conditions and PWB), and by nostalgia (which was also negatively related to home conditions and PWB).

To contrast H3 and H4, an MLG was performed. The model fits the data well (χ^2^_(96)_ = 272.19, GFI = 0.97, AGFI = 0.95, RMSEA = 0.03, TLI = 0.94, CFI = 0.96, NFI = 0.93, IFI = 0.96). As expected, from 45th day of confinement, both IA and nostalgia partially mediated the relationship between home conditions and PWB in a greater way (see Figure 1). Unexpectedly, there was no mediation effect before the 45th day of confinement.

### 3.2. Interaction between Sex and Days of Confinement

To contrast H5, 2 × 2 ANOVAs were conducted. As can be seen in Figure 2, the interaction effect is significant for both IA and nostalgia. As far as IA is concerned (F_(1, 1509)_ = 7.57, p = 0.006, ŋ^2 partial^ = 0.005), until the 44th day of confinement, men showed more levels of IA than women. From day 45th, the levels of addiction raised and were equal for both sexes, so women increased their levels of IA much more. In terms of nostalgia (F_(1, 1509)_ = 3.68, p = 0.05, ŋ^2 partial^ = 0.002), women scored higher in the two different time periods. There was a decrease in the levels for men and an increase for women from day 45.

## 4. Discussion

The measures taken by governments to fight the spread of COVID-19 meant a challenge for the populations, affecting their PWB. In this context, the levels of well-being may be affected by different circumstances. The general purpose of this study was to analyze the relationship of home conditions (i.e., having children, square meters of the house and square meters of the terrace or similar) with PWB and how this relationship could be mediated by IA and nostalgic feelings.

Firstly, and although it is not part of the objectives of this research, descriptive analyses showed an unexpected result, namely that the more children parents had, the more well-being they experienced. Traditionally, having children has been associated with lower parental well-being, mainly because of its effect on the economic sphere [33] or because of the stress that parenthood generates [34]. Other studies conducted during confinement also found a negative relationship [35]. However, other studies show that variables such as the type of lifestyle prior to parenthood [36] or the age of the children [37] could explain the existence of higher PWB. When the pre-parenting lifestyle was not very active and when the children were young, PWB would be higher. Future studies would have to consider these variables to determine their possible mediating role in the relationship between the number of children and parental PWB. It is likely that these variables, or others such as the sizes of the homes (they were larger when the participants in our study had more children), would provide a better understanding of our results.

Secondly, Internet use and its impact on psychological health have been studied thoroughly during the last two decades [38,39,40]. The excessive use of the Internet weakens social ties and reduces well-being levels [41,42], even reducing levels of attachment with close friends and family. Other studies show that certain use of the Internet can be beneficial to life satisfaction and to relationship-needs satisfaction [23]. Evidence has also been found that the use of the Internet can reduce perceived loneliness and increase self-esteem [43,44]. Therefore, the difference between moderate and excessive use of the Internet can lead to either positive or negative consequences. Our results suggest that those who lived in smaller places and with fewer children used the Internet more than those who lived in larger homes, with terraces and with more children, perhaps in the hope of being able to reduce the negative emotions they were experiencing, which led them to feel worse PWB. Thus, we can deduce that it is likely that many of them used the Internet in an excessive and compulsive way, and not in a healthy way.

On the other hand, it has been demonstrated that individuals feel nostalgia several times a week in usual conditions [20] independently from ages and cultures [22,24,45]. Although nostalgia has been associated with personal growth [21,23,45,46], or has been shown to be beneficial to psychological health [22,47,48,49,50], in stressful circumstances, a prevailing focus on the past or becoming excited about the future may induce negative effects on PWB [22]. Our results showed that those people living in larger places, with terraces and with more children experienced less nostalgia, probably because they had the opportunity to do activities that others in other conditions could not do. In addition, they also experienced better PWB, possibly because they were more focused on the future than in the present, which in previous research has been associated with more positive health behaviors [51]. We have not found previous research that has studied how home size, terrace size or the number of children can affect IA. The studies focused on the relationship with psychological, social, or personal factors [52,53,54], or on how parental Internet use and abuse can influence their children’s IA [55], probably because in the absence of confinement such conditions do not have much effect on the development of IA.

Another variable that can influence human processes is the time of confinement. In the case of IA, the evolution of Internet use may mean that people move from experiencing positive consequences, such as increased interpersonal relationships and subjective well-being [56,57], to experiencing different negative consequences associated with significant impairment and distress [58]. Our results showed that from the 45th day of confinement, people living in worse conditions experienced higher levels of IA. It is possible that these people increased their Internet use during the first 44 days of confinement to attend the time they previously spent on activities that could not be undertaken. When it was announced that the confinement measures were going to be relaxed, which meant that people could start to resume some activities, it is possible that some people perceived higher levels of IA because they were then aware of the interference of the Internet to do such activities. This could explain why from day 45 of confinement, the mediating effect of IA between home conditions and PWB was so strong.

Nostalgia can also be affected in different ways by the perceived temporal distance to certain events and self-evaluations [22]. When the government announced the relief of mobility restrictions, a large part of the population anticipated the activities in which they would be able to do again. The memory of these activities led to greater nostalgia towards them, which could explain why in our results the mediation of this variable had a greater impact from the 45th day of confinement than in the first 44 days, when no mediation effect of nostalgia was found. It is possible that for the first few weeks people perceived the real need to be confined to combat the pandemic, and accepted the reduction of freedom as temporary, which is the reason why the nostalgia did not affect their PWB.

Finally, our findings showed an interaction effect between confinement time and sex on both IA and nostalgia. As far as IA is concerned, previous research showed that men were at greater risk of developing IA than women [59,60,61,62]. Our results showed that during the first 44 days of confinement, men scored higher in IA than women confirming the results of previous research, but from day 45 the levels between both were similar. It seems that during the first weeks of confinement men took more refuge than women in the use of the Internet to cope with the consequences of isolation. Since the announcement of the relief measures, women especially reported higher levels of IA. Perhaps women tried to get more information about the progress of the pandemic to be better prepared for the “new normality”.

Regarding nostalgia, there is not a large bibliography about differences by sex, but most of the studies conclude that it serves psychological functions equally for women and men [20,24,26]. Our results provided evidence of a remarkable difference in the levels of nostalgia depending on the sex from day 45. Since that day, nostalgic feelings were enhanced in women, while they were decreased in men. The announcement of the relaxation of confinement had a stronger effect on women probably because they were more aware of the activities they could resume from that moment on. It is also possible that women would have had to adapt their roles more than men, ceasing to do more things than they did, and that for this reason the return to “normality” made them feel more nostalgic.

This work is not free of limitations, among which we can highlight the conditions under which the survey was administered. On the one hand, the questionnaire was completed online, with no monitoring of how it was answered. On the other hand, there was the possibility of overlapping with other questionnaires sent by other researchers, which could reduce the reliability and validity of the responses. To avoid people’s fatigue, our questionnaire gave clear instructions on the possibility of answering at different moments without losing the answers entered.

Furthermore, we assumed a unidirectional view of the relationships among the variables, when they can be bidirectional, that is, that PWB influence on IA or nostalgia. In future studies, it would be useful to develop longitudinal instead of cross-sectional designs in order to uncover reciprocal causal relationships.

Finally, we are aware of the exceptional circumstances of the COVID-19 quarantine, which was the first situation of its kind faced in modern history. Thus, both the behavior of the population and the measures taken by governments and health services could be different in successive pandemics.

## 5. Conclusions

In light of the above discussion and considering the consequences of COVID-19 and other previous diseases, implementing programs that promote healthy behaviors should be a priority for all governments, which endeavor to find innovative and successful treatment approaches [63]. In this sense, the more we know about variations in PWB and the factors that affect it in stressful situations of isolation, the more effective the health promotion strategies provided by the competent institutions will be.

The data from this study showed that home conditions are related to PWB, since people with better conditions in their homes kept their PWB at more positive levels. Improving people’s home conditions could also maintain low levels of IA (positive effect), whose high values affected negatively PWB. Similarly, better home conditions were also associated with lower levels of nostalgia, which resulted in higher levels of. PWB.

The relationships discussed in this document are sensitive to other environmental circumstances. Thus, we can observe that the interactions between the variables appeared from the day when a relief in the isolation measures was announced. Regarding IA, an increase in its values was shown from day 45 onwards, with this being greater in women than in men. In addition, a similar trend was observed related to feelings of nostalgia, although in this case, whereas women increased their values, men decreased them. In sight of these results, we can conclude that women were more sensitive to changes than men during confinement.

As quarantines are fortunately not common today, knowledge about the PWB of the population in the face of isolation measures is limited. Our findings present the effect of IA and nostalgia over the PWB state of the population in difficult times of isolation. Given the rise in the use of the Internet and the nostalgia feelings that usually emerge in restrictive situations, the results seem to be a useful tool to keep higher values of PWB. For this reason, our findings would be used by involved institutions to design interventions for keeping or improving the PWB of people in similar future situations, enhancing the positive response of people over the threat in a medium-large period of time.

## Figures and Tables

**Figure 1 ijerph-18-07386-f001:**
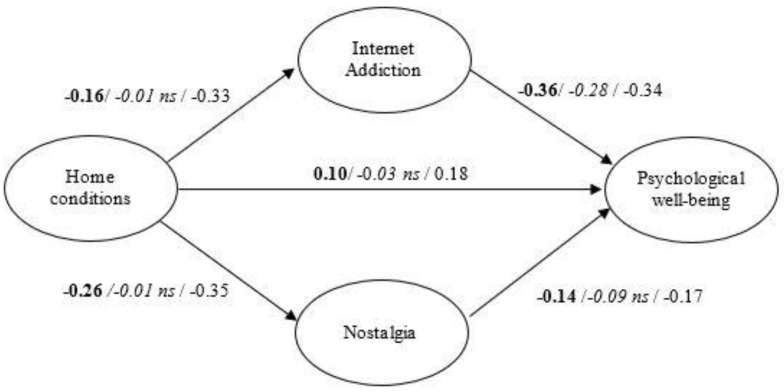
Research Model (**all**/*before 45 day*/from 45 day). ns = non-significant.

**Figure 2 ijerph-18-07386-f002:**
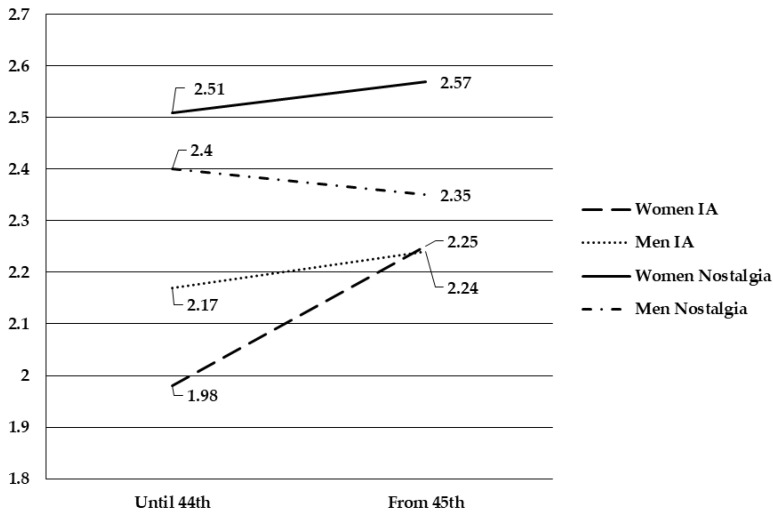
Interaction effects between sex and confinement in IA and nostalgia.

**Table 1 ijerph-18-07386-t001:** Home conditions. Means (M), Standard Deviations (SD), Medians (m) and correlations (N = 1509).

Dimension	Participants				Correlations							
	n	M	SD	m	(1)	(2)	(3)	(4)	(5)	(6)	(7)	(8)
1. Number of children	444	1.67	0.62	2	1	-	-	-	-	-	-	-
2. Square meters house	1509	107.36	70.35	90	0.89 **	1	-	-	-	-	-	-
3. Square meters terrace	744	108.01	647.18	19	0.05 *	0.15 **	1	-	-	-	-	-
4. IA	1509	2.16	0.69	2.1	−0.11 **	−0.01	0.01	1	-	-	-	-
5. Leisure-time activities	1509	2.62	0.65	2.57	−0.21 **	0.01	0.01	0.25 **	1	-	-	-
6. Intellectual activities	1509	1.99	0.75	2	−0.03	−0.02	0.05	0.13 **	0.35 **	1	-	-
7. Basic activities	1509	2.65	0.66	2.67	0.02	−0.06 *	−0.05	0.01	0.15 **	0.21 **	1	-
8. PWB	1509	2.84	0.78	2.91	0.13 **	0.02	0.03	−0.34 **	−0.18 **	−0.14 **	−0.05	1

Notes. n = number of respondents. * *p* < 0.05, ** *p* < 0.01.

## Data Availability

The datasets generated and/or analyzed during the current study are not publicly available but are available from the corresponding author on reasonable request.

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
