# Peer review of "Psychological Well-Being and Home Conditions during COVID-19 Confinement. Internet Addiction and Nostalgia as Mediators"

_ijerph, 2021, doi:10.3390/ijerph18147386_

Round 1

Reviewer 1 Report

The manuscript is Libano et al study aimed “to determine if home conditions (i.e., having children, square meters of the house and square meters of the terrace or similar) relate to the psychological wellbeing of people in confinement and if this relationship is mediated by Internet Addiction and by nostalgia”.

While the concept is interesting, and the findings add to our understanding of the consequence of confinement, there are some issues need to be addressed.

1. Authors have any idea why the proportion of women accepted to participate was higher than men? Given that housewives relatively spent more time at home before covid pandemic, how many of participants were housewives?

2. Were there any effects and interactive effects for age and employment status? 3. Why didn’t the authors consider the marital status?

4. How did the authors calculate home conditions from number of children, square meter of the house and terrace (figure 1)?

5. Section 3.2 and figure 2 might mislead readers that the study was a longitudinal study and any given participants have been evaluated at two consecutive timepoints.

Author Response

Dear reviewers:

First, we would like to thank you for reviewing our research. We are convinced that based on your suggestions we have considerably improved the manuscript. We have tried to follow all your suggestions.

To make it easier to locate changes in the manuscript, we have assigned each of you a different color. This color can be found at the beginning of each reviewer's suggestions.

In addition, we have taken advantage to correct some typos that we have found on the text.

Response to Reviewer 1 Comments

Point 1: The manuscript is Libano et al study aimed “to determine if home conditions (i.e., having children, square meters of the house and square meters of the terrace or similar) relate to the psychological wellbeing of people in confinement and if this relationship is mediated by Internet Addiction and by nostalgia”.

While the concept is interesting, and the findings add to our understanding of the consequence of confinement, there are some issues need to be addressed.

Authors have any idea why the proportion of women accepted to participate was higher than men? Given that housewives relatively spent more time at home before covid pandemic, how many of participants were housewives?

Response 1: Although further research may be needed, according to Smith (2008), we can predict a higher number of responses from women due to the questionnaire used in this research was based on “connective” variables, such as Psychological Well-Being (PWB) and nostalgia. Apparently, there are some differences by gender in the disposition to collaborate regarding the prevailing topic in the study. Thus, it is more likely to get responses from men when the topic is related to “separative” characteristics (e.g. preferences and opinions about tangible stuff), while women usually present a higher percentage of participation in those studies focused on “connective” variables.

When it comes to the second question, the aim of our study was focused on the effect of different variables related to home conditions on the PWB. According to the literature reviewed, we opted for the variables like the number of people sharing a flat, the number of children at home, and the space of their homes. The essence of the special restrictions during the lockdown made us understand these were the conditions to study. Thus, no one question about the job position was included in the survey. Within the sociodemographic section, we did include the type of occupation, where two answers might be proper for housewives (“Unemployed” and “Other type”) but it is impossible to know the occupation itself. What we can assure is that the unemployed rate of women during lockdown was approximately 2,59% higher in women than in men (16,72%-14,13%, respectively; INE, 2020). We really think that this issue might deserve a whole study to analyse how was the PWB situation for housewives specifically. On this regard and according to the study of Farre et al. (2020), it seems that women assumed disproportionately most of the household than men before and during the lockdown, and sounds logical that it has an effect on PWB, but the origin of this matter is not include in the home conditions variables of our study.

Point 2: Were there any effects and interactive effects for age and employment status?

Response 2: In our research we assessed the age variable quantitatively, so we did not grouped data. Moreover, as testing the effect of age was not an objective of our research, we did not search the literature to see if being in a particular age group could influence our dependent variables differently. For this reason, we have not included the effects of age on our results. Following your suggestion, we have tested the relationship between age and internet addiction, and age and homesickness, and the relationship in both cases is negative and significant (p<.001). That is, the higher the age, the lower the levels of Internet addiction and the lower the levels of homesickness.

As far as employment status is concerned, it was neither part of our research objectives to find out its effect on Internet addiction and nostalgia. For this reason, we did not include it in the manuscript. Following your suggestion, we have analysed its effect, and, in both cases, there are statistically significant differences. Both in the case of Internet addiction and nostalgia, the group with the lowest levels was the unemployed, the retired and those on sick leave (n=173). The group with the highest levels of Internet addiction and  nostalgia was composed of those who were studying and working at the same time, although at the time of the research they were not working because of covid-19 (n=386).

Point 3: Why didn’t the authors consider the marital status?

Response 3: Although marital status may indeed be an interesting variable that may have different effects on the dependent variables of our research, the previous literature review led us to select other variables such as the number of people sharing a flat, the number of children they had, or the space of their homes as independent variables. If similar conditions are repeated in the future, it would be advisable to include marital status in the research design.

Point 4: How did the authors calculate home conditions from number of children, square meter of the house and terrace (figure 1)?

Response 4: As briefly described in lines 131-133, the number of children, the square meters of the house and the square meters of the terrace, balcony or similar, were asked through open questions. To include these variables in the structural equation model, we associate them with a latent variable that we call home conditions. To include these variables in the structural equation model, we associate them with a latent variable that we call home conditions. Figure 1 represents the model in a simple way, as including all observable variables would probably impair the reader's understanding.

Point 5: Section 3.2 and figure 2 might mislead readers that the study was a longitudinal study and any given participants have been evaluated at two consecutive timepoints.

Response 5: We agree with the reviewer's appreciation when referring to figure 2. Indeed, it could be understood that this is a longitudinal study. However, in the procedure section we indicate that it is a cross-sectional study, and, in the discussion, we include this as a limitation of the study, recommending that longitudinal studies be carried out in future research. Finally, the type of statistical analysis we performed to test the hypothesis 5, i.e., 2x2 ANOVAs (included in section 3.2), is a type of analysis that is usually used in cross-sectional designs, while repeated measures ANOVAs are usually used in longitudinal designs. For all these reasons, we believe that it is preferable to keep figure 2 in the manuscript, as in our opinion it allows a better understanding of the results obtained.

Reviewer 2 Report

The manuscript "Psychological well-being and home conditions during COVID-19 confinement. Internet addiction and nostalgia as mediators." by Del Líbano et al is well written and interesting. Minor comments regarding this present work are:

*The evaluated home conditions are listed, but "better" home conditions are never defined. Do the authors mean more squaremeters, less children, etc.? 

*The correlation e.g. between number of children and PWB is mentioned in the results but not discussed. Please also briefly discuss your side findings, as they are partially surprising.

*Did the authors evaluate income and social status? This might be relevant in this context. The mean size of house (107m²) and terrace (53m²) areas appear very generous, but maybe they are standard for Spain and/or the particular living area. Please shortly elaborate.

*Introduction: (Line 42) Did the authors mean "While quarantine has never been seen before by most of the world's population"?

*Discussion: (Line 236) Did the authors mean "independent from ages and cultures"

Author Response

Dear reviewers:

First, we would like to thank you for reviewing our research. We are convinced that based on your suggestions we have considerably improved the manuscript. We have tried to follow all your suggestions.

To make it easier to locate changes in the manuscript, we have assigned each of you a different color. This color can be found at the beginning of each reviewer's suggestions.

In addition, we have taken advantage to correct some typos that we have found on the text.

Response to Reviewer 2 Comments

Point 1: The manuscript "Psychological well-being and home conditions during COVID-19 confinement. Internet addiction and nostalgia as mediators." by Del Líbano et al is well written and interesting. Minor comments regarding this present work are:

The evaluated home conditions are listed, but "better" home conditions are never defined. Do the authors mean more squaremeters, less children, etc.?

Response 1: We totally agree with the reviewer on this comment. The use of the word “better” is inaccurate and may be confusing. There is not a better condition over others, because the objective of the paper was to set down a relation about home condition and PWB, but this is not enough to consider a categorical term. On this ground, we have modified the text to be more accurate with the terminology (lines 247-260).

Point 2: The correlation e.g. between number of children and PWB is mentioned in the results but not discussed. Please also briefly discuss your side findings, as they are partially surprising.

Response 2: Following the reviewer's suggestion, we have included a paragraph at the beginning of the discussion (lines 227-239)  in which we refer to this unexpected result and discuss possible explanations for it.

Point 3: Did the authors evaluate income and social status? This might be relevant in this context.

Response 3: The answer to the reviewer's question is no, we did not ask about income or social status. Although we also considered the possibility of evaluating them, since both variables probably influenced the psychological well-being of individuals, their levels of Internet addiction and nostalgia, given that they are data that can be influenced by social desirability, we finally preferred to focus on other types of information.

Point 4: The mean size of house (107m²) and terrace (53m²) areas appear very generous, but maybe they are standard for Spain and/or the particular living area. Please shortly elaborate.

Response 4: Following the reviewer's suggestion, we have improved the information shown in Table 1. Specifically, we have included the persons who responded to each variable to better appreciate the most common living conditions in Spain. In addition, we have also included the median together with the mean and standard deviation, to facilitate the understanding of the information. Finally, since we have considered only those people who had children or had a terrace to represent the number of children and the square meters of the terrace, the means and standard deviations of these variables have changed.

Point 5: Introduction: (line 42) Did the authors mean "While quarantine has never been seen before by most of the world's population"?

Response 5: That is exactly what we meant. We really appreciate the correction. The text has been modified (line 46).

Point 6: Discussion: (line 236) Did the authors mean "independent from ages and cultures"

Response 6: As said in the previous point, we appreciate again the correction and have modified the text according the suggestion (line 254).

Reviewer 3 Report

The paper is well written, generally respecting the review's guidelines. Nevertheless, I consider the research's innovative character to be questionable, and the authors should better explore the practical implications of the data obtained. Considerations are made about the importance of home conditions in psychological well-being, but there is no development on IA, considering that technology is today an essential tool to keep people connected with the rest of the world.

Author Response

Dear reviewers:

First, we would like to thank you for reviewing our research. We are convinced that based on your suggestions we have considerably improved the manuscript. We have tried to follow all your suggestions.

To make it easier to locate changes in the manuscript, we have assigned each of you a different color. This color can be found at the beginning of each reviewer's suggestions.

In addition, we have taken advantage to correct some typos that we have found on the text.

Response to Reviewer 3 Comments

Point 1: The paper is well written, generally respecting the review's guidelines. Nevertheless, I consider the research's innovative character to be questionable, and the authors should better explore the practical implications of the data obtained.

Response 1:. The prevailing aim of this paper is to analyse what variables (some of them) related to citizen’s PWB we must take in consideration when population is in isolated and threatened situation to keep good levels of PWB. According to the initial purpose of the project that gave rise to this study, we have added some information about the utility of our results (lines 345-351).

Point 2: Considerations are made about the importance of home conditions in psychological well-being, but there is no development on IA, considering that technology is today an essential tool to keep people connected with the rest of the world.

Response 2: The reviewer's suggestion is particularly interesting. We did not discuss the influence of conditions on internet addiction because we did not find research that had previously studied such conditions. In order to provide more information on this issue, we have briefly discussed the result on lines 263-267 below.

Round 2

Reviewer 1 Report

I have no more questions.